# 5-Aminolevulinic Acid Induces Chromium [Cr(VI)] Tolerance in Tomatoes by Alleviating Oxidative Damage and Protecting Photosystem II: A Mechanistic Approach

**DOI:** 10.3390/plants12030502

**Published:** 2023-01-21

**Authors:** Cengiz Kaya, Ferhat Ugurlar, Muhammed Ashraf, Mohammed Nasser Alyemeni, Michael Moustakas, Parvaiz Ahmad

**Affiliations:** 1Soil Science and Plant Nutrition Department, Harran University, 63200 Sanliurfa, Turkey; 2Institute of Molecular Biology and Biotechnology, The University of Lahore, Lahore 54600, Pakistan; 3Botany and Microbiology Department, King Saud University, Riyadh 11451, Saudi Arabia; 4Department of Botany, Aristotle University of Thessaloniki, 54124 Thessaloniki, Greece; 5Department of Botany, GDC, Jammu and Kashmir, Pulwama 192301, India

**Keywords:** chromium sequestration, sodium nitroprusside, sodium tungstate, hydrogen peroxide, electrolyte leakage, phytochelatins, nitrate reductase, nitric oxide, antioxidant enzymes

## Abstract

Chromium [Cr(VI)] pollution is a major environmental risk, reducing crop yields. 5-Aminolevunic acid (5-ALA) considerably improves plant abiotic stress tolerance by inducing hydrogen peroxide (H_2_O_2_) and nitric oxide (NO) signalling. Our investigation aimed to uncover the mechanism of tomato tolerance to Cr(VI) toxicity through the foliar application of 5-ALA for three days, fifteen days before Cr treatment. Chromium alone decreased plant biomass and photosynthetic pigments, but increased oxidative stress markers, i.e., H_2_O_2_ and lipid peroxidation (as MDA equivalent). Electrolyte leakage (EL), NO, nitrate reductase (NR), phytochelatins (PCs), glutathione (GSH), and enzymatic and non-enzymatic antioxidants were also increased. Foliar application of 5-ALA before Cr treatment improved plant growth and photosynthetic pigments, diminished H_2_O_2_, MDA content, and EL, and resulted in additional enhancements of enzymatic and non-enzymatic antioxidants, NR activity, and NO synthesis. In Cr-treated tomato seedlings, 5-ALA enhanced GSH and PCs, which modulated Cr sequestration to make it nontoxic. 5-ALA-induced Cr tolerance was further enhanced by sodium nitroprusside (SNP), a NO donor. When sodium tungstate (ST), a NR inhibitor, was supplied together with 5-ALA to Cr-treated plants, it eliminated the beneficial effects of 5-ALA by decreasing NR activity and NO synthesis, while the addition of SNP inverted the adverse effects of ST. We conclude that the mechanism by which 5-ALA induced Cr tolerance in tomato seedlings is mediated by NR-generated NO. Thus, NR and NO are twin players, reducing Cr toxicity in tomato plants via antioxidant signalling cascades.

## 1. Introduction

Chromium (Cr) is one of the most toxic heavy metals found naturally and broadly used in industrial processes, such as the manufacturing of chemicals, chrome plating, alloys, paints, various pesticides, fertilizers, leather tanning, and metallurgical and other works [1,2]. Humans can be exposed to Cr toxicity in various ways, such as the plant–soil–water system and from drinking Cr-contaminated water [3]. Chromium in soil and in irrigation water is taken up by plants and restricts the growth and yield of crops and, furthermore, this can be detrimental to human health as it accesses the food chain [2,4]. Chromium is present in several forms in soil, but the well-known and constant forms are Cr(III) and Cr(VI), with Cr(VI) having greater capability of penetrating plant roots [2]. The hexavalent form, Cr(VI), by entering plant roots, can disrupt crucial cellular structures more than other Cr forms [5,6]. The form Cr(III) is required for humans, while the Cr(VI) form is reported as a human health hazard [2]. Chromium toxicity can oxidize plant cell biomolecules due to the over generation of reactive oxygen species (ROS), such as hydroxyl radical (OH^•^), singlet oxygen (^1^O_2_), and hydrogen peroxide (H_2_O_2_) [7,8]. An antioxidant defense system is developed by plants to lessen the oxidative damage due to Cr toxicity [9]. This defense mechanism includes the synthesis of non-enzymatic substances such as glutathione (GSH), which is involved in the enhancement of antioxidant enzyme activities [10,11]. Sequestration of toxic metals in vacuoles by phytochelatins (PCs) is another important defense system for plant tolerance to toxic metals [12], which can be stimulated to efficiently mitigate the adverse effects of Cr toxicity. However, such protective pathways against Cr toxicity are not often active in various plant species such as tomatoes [13].

5-aminolevulinic acid (5-ALA) is one of the bio-stimulants involved in the plant defense system against biotic and abiotic stresses [14,15], and it might possess an outstanding role in enhancing plant tolerance to Cr toxicity. 5-ALA is a precursor of all porphyrin compounds found in bacteria, fungi, and plants, as well as a key plant growth stimulator against several metal toxicities [15,16,17]. A stimulating effect of 5-ALA on metabolic and photosynthesis events in several plant species has been observed [18]. Supplementation of 5-ALA has previously been shown to enhance the tolerance of cucumbers to salinity and low-temperature stress [19,20], and of strawberries to osmotic stress [21]. It has been suggested that 5-ALA improves plant abiotic stress tolerance by inducing hydrogen peroxide (H_2_O_2_) and nitric oxide (NO) signalling [22]. Recently, foliar application of 5-ALA has been suggested as a feasible alternative approach to alleviate Cr-induced stress in crop plants, but reports on the beneficial effect of 5-ALA on Cr stress in plants are rare [23]. It has been shown that 5-ALA decreased Cr-induced oxidative stress in maize by improving the antioxidative enzyme activities of peroxidase (POD), superoxide dismutase (SOD), and catalase (CAT), and enhanced tolerance to Cr toxicity by maintaining optimal plant growth and positive photosynthetic procedures [23]. Strawberry tolerance to osmotic stress has been shown to be mediated by ABA signalling and H_2_O_2_ accumulation after application of 5-ALA [21]. However, the mechanism by which 5-ALA induced Cr tolerance in crop plants remains to be elucidated.

Nitric oxide (NO) has been revealed to function as a cell signalling molecule involved in hormonal signalling, developmental processes, and acclimation response to environmental stresses [24]. NO generated in plants can alleviate oxidative stress by inducing post-translational modifications in the enzymes/proteins involved in antioxidant defence responses [25]. NO is known to scavenge H_2_O_2_, thereby protecting plant cells from ROS-mediated damage [26,27,28]. An increase in the H_2_O_2_ content in leaf tissues results in the oxidation of macromolecules, proteins, lipids, and DNA [29,30]. The enzyme NR is responsible for the generation of NO in plants under stressful environmental conditions [31,32]. NO is a signal molecule with a constructive function in seed germination [33], adventitious rooting [34], phytohormone homeostasis and perception [35], and fruit ripening [36], as well as a regulating role in the defense systems of plants under several stresses including arsenic stress [37], cadmium stress [38], salinity stress [39], and drought stress [40].

In addition to the well-known protective roles of 5-ALA and NO in plant tolerance against several stresses, the collaborative role of 5-ALA and NO in Cr-treated tomato seedlings has not yet been examined in detail. Our hypothesis was that 5-ALA and NO may alleviate Cr-induced toxicity in tomato plants under Cr treatment. Thus, this investigation was intended to reveal the mechanism by which 5-ALA can induce Cr tolerance in tomato plants, with or without the NR inhibitor sodium tungstate (ST) and/or with the NO donor sodium nitroprusside (SNP). Accordingly, we examined (i) the potential roles of NR and NO in alleviating oxidative damage and protecting photosystem II (PSII), (ii) the subcellular Cr distribution and Cr sequestration for detoxification, and (iii) the involvement of enzymatic and non-enzymatic antioxidants in the 5-ALA-induced Cr tolerance of tomato seedlings.

## 2. Results

### 2.1. Chromium and 5-ALA Treatments on Phenotypic Appearance of Tomato Seedlings

The effects on tomato leaves growing under Cr stress and supplemented with 5-ALA alone or together with sodium tungstate (ST), or combined with sodium nitroprusside (SNP), a NO donor, are illustrated in Figure 1. Yellowing symptoms appeared on the leaves of tomato seedlings under Cr stress. Both leaf size and seedling height decreased in Cr-stressed seedlings, while those sprayed with 5-ALA did not exhibit those disorders on leaves. However, the pre-treatment with ST, an inhibitor of NR, inverted the improved phenotypic appearance by 5-ALA, while the application of SNP, a NO donor, mitigated the negative effect of ST (Figure 1).

### 2.2. Chromium and 5-ALA Treatments on Tomato Biomass

Chromium (Cr)-treated tomato seedlings displayed substantial decreases in shoot, root, and total dry weights by 25.6%, 38.6%, and 28.5%, respectively, compared to untreated plants (Figure 2A–C). In contrast, 5-ALA application in plants before Cr treatment resulted in substantial increases in the above-mentioned parameters, by 17.2%, 50.0%, and 22.9%, respectively, compared to seedlings treated with Cr alone (Figure 2A–C). The addition of SNP to 5-ALA (+Cr + 5-ALA + SNP) resulted in further increases in these traits over plants treated with Cr alone, and the data were not significantly different from control plants, indicating that 5-ALA + SNP positively affected the growth of Cr-treated plants (Figure 2A–C). The supplementation of ST (NR inhibitor) to 5-ALA completely eliminated the alleviation effect of 5-ALA on Cr-treated seedlings, revealing that NR contributes to 5-ALA enhancement of plant growth. Moreover, these data suggest that when plants are pre-treated with ST, 5-ALA supplementation does not effectively mitigate the harmful effects of Cr stress due to the inhibited NR activity and endogenous NO. The treatment of 5-ALA + ST + SNP overturned the adverse effect of ST, indicating that endogenous NO is needed for the 5-ALA enhancement of plant growth under Cr toxicity. In control plants, these parameters did not show a substantial difference under all treatments, suggesting that 5-ALA or ST are not efficient in modulating plant growth under non-stress conditions (Figure 2A–C).

### 2.3. Chromium and 5-ALA Treatments on Photosynthesis-Associated Traits

Chromium (Cr)-treated tomato seedlings showed a significant (*p* ≤ 0.05) decrease in chlorophyll *a,* chlorophyll *b* content, and photosystem II maximum efficiency (F*v*/F*m*), by 43%, 20%, and 25%, respectively, compared to control plants (Figure 2D–F). The F*v*/F*m* value of control leaves of tomato seedlings sprayed with the surfactant solution (0.01% Tween-20) was 0.799 ± 0.009 (Figure 2F), which is 4% lower than those that were not treated with Tween-20 (0.831 ± 0.003) (data not shown). Since the difference was not significant, we used those with Tween-20 as control values (Figure 2F), so that the difference would reflect ALA treatment alone. 5-ALA application in plants before Cr treatment increased chlorophyll *a,* chlorophyll *b* content, and photosystem II maximum efficiency by 36%, 13%, and 33%, respectively (Figure 2D–F). Supplementation of SNP along with 5-ALA in Cr-treated seedlings resulted in further enhancement of chlorophyll *a* and chlorophyll *b* content, which does not differ from controls (Figure 2D,E), while it did not have any positive or negative effect on F*v*/F*m* (Figure 2F). The pre-treatment of ST reversed the beneficial impact of 5-ALA on chlorophyll *a* and chlorophyll *b* content, suggesting that elevated NR activity is needed for 5-ALA to be effective for the enhancement of chlorophyll *a* and chlorophyll *b* content, but not for the enhancement of photosystem II maximum efficiency (F*v*/F*m*) (Figure 2F). The supplementation of SNP along with 5-ALA under Cr stress eliminated the adverse impact of ST on chlorophyll content by increasing NO. All treatments of control plants did not show any negative changes in photosynthesis-related traits (Figure 2D–F), suggesting that no harmful effect of ST or no positive effect of 5-ALA were observed on photosynthesis-linked attributes in the unstressed plants.

### 2.4. Chromium and 5-ALA Treatments on Leaf–Water Relations and Proline Content

Seedlings treated with Cr showed significant (*p* ≤ 0.05) reductions in leaf relative water contents (RWC; 30.9%) and water potential (ΨI; 2.1-fold), whereas a considerable elevation in proline content (5.2-fold) compared to controls was noticed (Figure 3A–C).

However, 5-ALA application in Cr-treated seedlings enhanced all the above parameters compared to those of treated with Cr alone (Figure 3A–C). However, SNP supplement to 5-ALA (5-ALA + SNP) resulted in further increases in RWC and ΨI (Figure 3A,B), and these data were not different from those obtained in control seedlings, except for the leaf water potential (ΨI), which was lower than that of the control. Although the beneficial effects of 5-ALA on these attributes were reversed by ST supplementation, those of 5-ALA + SNP + ST were not inverted to the same extent, suggesting that NR and NO collaborated in improving 5-ALA-induced improvement in water-relation traits (Figure 3A–C). Various treatments with control plants did not alter these parameters, suggesting no beneficial or harmful effect of 5-ALA or ST on non-stressed tomato seedlings (Figure 3A–C).

### 2.5. Chromium and 5-ALA Treatments on Nitrate Reductase (NR) Activity and Nitric Oxide (NO) Synthesis

To obtain a possible insight into the participation of NR in 5-ALA-enhanced Cr tolerance of tomato plants, alterations in NR activity and NO synthesis along with leaf 5-ALA content were also estimated. Chromium-treated plants exhibited a slight non-significant decrease in NR activity (17.1%) and NO synthesis (16.3%), but 5-ALA content increased significantly (81.2%) compared to controls (Figure 3D–F).

The supplementation of 5-ALA and 5-ALA + SNP resulted in the elevation of NR activity, NO synthesis, and 5-ALA content in Cr-treated tomato seedlings (Figure 3D–F), while the pre-treatment with ST reversed these elevations (Figure 3D–F), but did not change 5-ALA content (Figure 3D–F). However, the addition of SNP along with 5-ALA eliminated the adverse effect of ST on NR activity and NO synthesis (Figure 3D,E). 

### 2.6. Chromium and 5-ALA Treatments on Oxidative Stress Markers

Oxidative stress-related markers were significantly (*p* ≤ 0.05) increased under Cr stress. Hydrogen peroxide (H_2_O_2_) increased 3.6-fold, lipid peroxidation (as MDA) 3.5-fold, and electrolyte leakage (EL) 2.4-fold (Figure 4A–C). 

Application of 5-ALA to Cr-treated seedlings reduced H_2_O_2_, MDA, and EL, by 41%, 31%, and 33%, respectively, compared to Cr-treated seedlings alone. Moreover, the pre-treatment of 5-ALA together with SNP led to additional decreases in oxidative stress markers in Cr-treated seedlings. However, the ST treatment that decreased NR activity and NO synthesis (Figure 3D,E) reversed the positive impact of 5-ALA applied alone, and that of 5-ALA plus SNP (Figure 4A–C). 

### 2.7. 5-ALA Regulated Subcellular Chromium Content in Tomatoes

Chromium-treated seedlings showed remarkably (*p* ≤ 0.05) higher cellular Cr accumulation in root (Figure 5A) and leaf (Figure 5B) tissues. Cr was not detected in the leaves and roots of control seedlings because of a lack of Cr in the root zone. Treatment with 5-ALA elevated cell wall Cr fraction in the roots. The supplementation of SNP along with 5-ALA (5-ALA + SNP) led to a further increase in the cell wall Cr fraction. These findings clearly indicate that 5-ALA promotes the cell wall Cr fraction in root cells, thereby reducing its toxic effect within the cell. In the leaves of Cr-treated plants, Cr was deposited largely in the vacuoles as soluble fraction (56%), followed by the cell wall fraction (38%) and cell organelles (6%) (Figure 5B).

The supplementation of 5-ALA and 5-ALA + SNP resulted in a decrease in the cell wall leaf Cr fraction and in an increase in the vacuole soluble Cr fraction in the leaves of Cr-treated seedlings. Conversely, ST pre-treatment prevented this controlling influence of 5-ALA. In light of these results, the beneficial effect of 5-ALA on Cr-stressed plants might have been caused by an increase in the activity of NR and NO synthesis. The NO produced by NR could be a downstream molecular signal of 5-ALA-induced Cr tolerance in tomato seedlings.

### 2.8. Phytochelatin, Glutathione, and Ascorbate Synthesis under Chromium and 5-ALA Treatments

Chromium-treated seedlings showed significant elevations in phytochelatin synthesis (PC, 6.5-fold) and the glutathione *S*-transferase (GST, 1.6-fold) activity (Figure 6A,B). Moreover, they showed increases in reduced glutathione (GSH, 56%) and oxidized glutathione (GSSG, 2-fold) content, but reduction in the ratio of GSH/GSSG (Figure 6C–E)compared to control plants. The supply of 5-ALA along with Cr treatment resulted in increasing all these parameters, except the GSSG. The treatment of 5-ALA plus SNP resulted in further enhancement of these traits, also except the GSSG. These data suggest that 5-ALA alone or jointly with SNP may result in detoxification of Cr by boosting PC and GSH synthesis.

There were significant reductions in ascorbate (AsA, 23%), but an elevation in dehydroascorbate (DHA, 34%) levels in Cr-treated plants compared to control plants (Figure 7A,B). Consequently, the rate of AsA/DHA was diminished in Cr-treated plants by 42% over that of the control (Figure 7C). In Cr-treated plants, AsA levels remained unchanged with 5-ALA treatment, but increased with 5-ALA + SNP (Figure 7A), while DHA remain unchanged under both treatments (5-ALA and 5-ALA + SNP) (Figure 7B). The ratio of AsA/DHA remained unchanged with 5-ALA treatment but increased with 5-ALA + SNP in Cr-treated plants (Figure 7C).

Sodium tungstate (ST) treatment inverted the impact of 5-ALA on all attributes mentioned above, but the supply of 5-ALA along with SNP reversed the negative effect of ST on all parameters in Cr-treated plants.

### 2.9. Ascorbate-Glutathione Cycle Enzymes under Chromium and 5-ALA Treatments

We measured the ascorbate-glutathione (AsA-GSH) cycle-associated enzymes, which are glutathione reductase (GR), ascorbate peroxidase (APX), monodehydroascorbate reductase (MDAR), and dehydro-ascorbate reductase (DHAR) (Figure 7D–G). Chromium treatment elevated (*p* ≤ 0.05) the APX, GR, MDHAR, and DHAR activities 2.3-, 1.7-, 1.7-, and 1.6-fold, respectively, compared to those in the control seedlings (Figure 7D–G). The supply of 5-ALA or 5-ALA + SNP further elevated all these enzymes’ activities in Cr-treated plants (Figure 7D–G).

The pre-treatment of ST along with 5-ALA (5-ALA + ST*)* in Cr-treated plants abolished the positive influence of 5-ALA on these enzymes’ activities, possibly by down-regulating the activity of NR and NO synthesis. This establishes the fact that 5-ALA requires NR and NO to boost the AsA-GSH cycle. The adverse impact of ST on these enzymes’ activities was eliminated by SNP supplementation along with 5-ALA (5-ALA + SNP + ST) (Figure 7D–G).

## 3. Discussion

Our data indicated that chromium toxicity caused a decline in plant growth, as was previously observed in chickpeas [41], sunflowers [42], and maize [43]. This plant growth reduction that we observed was due to water uptake limitation by Cr, which led to low plant water content [44,45]. Application of 5-ALA mitigated the negative effect of Cr on plant growth, thus it can be suggested as an efficient growth promoter that contributes to Cr tolerance, as observed previously in maize [23] and sunflowers [46]. This positive effect of 5-ALA on plant growth in Cr-treated tomato seedlings might have been related to the improved photosynthetic function [46,47].

When tomato seedlings were pre-treated with ST before Cr treatment, foliar application of 5-ALA was not efficient in improving tomato growth due to the inhibition of NR activity and NO synthesis. Therefore, 5-ALA and NR collaborate to enhance growth under Cr stress. Moreover, our data show that NO synthesis was also eliminated by the inhibition of NR activity, indicating that the key source of NO synthesis under Cr stress is NR, and NO may be the downstream signal molecule. In order to have more evidence that NO contributes to 5-ALA-induced Cr tolerance in tomato seedlings, SNP was provided with 5-ALA plus ST. The data imply that SNP treatment inverted the harmful impact of ST on the evaluated parameters under Cr stress and suggest that NO contributed to the 5-ALA-induced Cr tolerance of tomato seedlings. Thus, 5-ALA and NO function jointly to boost the Cr tolerance of tomato plants and, thus, the beneficial effect of 5-ALA is dependent on NO. The mitigation of Cr toxicity by NO applied as SNP has been studied earlier in tomato and maize plants [48,49].

The reductions in chlorophyll content and F*v*/F*m* values that we observed in Cr-treated tomato seedlings (Figure 2D–F) were attributed to the increased H_2_O_2_ generation (Figure 4A) [50]. However, 5-ALA application restored F*v*/F*m* values (Figure 2F) with a parallel decrease in H_2_O_2_ accumulation (Figure 4A) in Cr-treated tomato plants, suggesting that the 5-ALA signaling cascade participates in alleviating the detrimental impact of Cr toxicity on photosynthetic function by eliminating H_2_O_2_ production. 5-ALA supplementation enhances chlorophyll content, as well as other photosynthetic parameters, since it is a precursor of chlorophyll biosynthesis [18,19].

Chromium toxicity results in enhancement of proline content in cereal species [51], as we observed in tomatoes. Plants subjected to drought or salt stress boost proline content to enhance water uptake and improve stress tolerance, since proline also functions as a protective agent against ROS [52,53]. However, the increased proline content in Cr-treated tomato plants was not enough to scavenge H_2_O_2_ and lipid peroxidation in order to alleviate oxidative damage. Nevertheless, 5-ALA enhanced proline synthesis and/or reduced proline catabolism in Cr-treated tomatoes, and this might have improved leaf water content and reduced the oxidative impairment. 5-ALA has been reported to enhance proline content in salt-stressed rapeseed [54] and in water-stressed sunflowers [55]. The beneficial effect of 5-ALA on proline content and leaf water status in Cr-treated tomato plants was eliminated by ST pre-treatment, indicating that 5-ALA enhancement of NR activity improved leaf water status and proline content, likely by generating NO. A positive influence of NO on water content has already been documented in tomatoes [56] and maize [49] under Cr stress.

The reductions in NR activity and NO synthesis in Cr-stressed tomato plants that we observed in our experiments have also been reported recently [57]. The supplementation of 5-ALA resulted in an enhancement of NR activity and NO synthesis in Cr-treated tomatoes, revealing that NR induced NO synthesis as a downstream biomolecule signal in plants treated with 5-ALA. This NO signaling can possibly explain the up-regulation of the AsA-GSH cycle by 5-ALA, which decreased the oxidative stress induced by Cr toxicity, thereby resulting in enhanced Cr tolerance of tomatoes. A reasonable level of NO synthesis has already been stated to improve plant tolerance to biotic and abiotic stresses [58]. In our study, the amount of NO stimulated by 5-ALA should have been at the range of nontoxic doses leading to nontoxic plant metabolic events in tomato seedlings. 5-ALA-generated NO synthesis enhancing tolerance has been reported under salinity stress for strawberry [22], and maize plants [59], but no data exist for Cr stress. 5-ALA enhances plant growth and yield, as well as abiotic stress tolerance, by regulating photosynthetic and antioxidant mechanisms and plant nutrient uptake [60]. However, it seems that its regulatory role is primarily focused on the physiological effects rather than the molecular mechanisms [60]. Assessing our data, it becomes evident that 5-ALA induced NO synthesis by triggering NR, but ST pre-treatment along with 5-ALA eliminated the positive impact of 5-ALA through inhibition of the activity of NR and NO synthesis. Correspondingly, 5-ALA has been reported to enhance NR activity in watermelon seedlings under salt stress [61] and UV-B-challenged *Cajanus cajan* L. plants [62]. Moreover, the inhibitory effect of ST on NR activity has previously been established under different stresses [31,63,64].

5-ALA treatment reduced the subcellular root and leaf Cr content under Cr stress in tomato seedlings, suggesting that 5-ALA can be used to enhance the Cr tolerance of tomato plants. Noticeably, the pre-treatment of tomato seedlings with ST reduced NR activity and NO synthesis and inverted the positive effect of 5-ALA that promoted the cell wall Cr fraction in root cells, thereby reducing its toxic effect within the cell.

We observed an excess accumulation of H_2_O_2_ and MDA under Cr stress, as was reported previously [65]. Cr-induced H_2_O_2_ and MDA accumulation leads to higher EL [47], as we also noticed. 5-ALA or 5-ALA + SNP mitigated the harmful impact of oxidative damage due to Cr stress in tomato seedlings by decreasing MDA and H_2_O_2_ contents. The promising effect of SNP supply as a donor of NO has also been examined in maize [49]. The protective role of 5-ALA or SNP under Cr stress was reported previously in tomatoes [66] and sunflowers [47]. Conversely, the supplementation of ST deactivated NR activity and NO synthesis, thereby reversing the protective function of 5-ALA on oxidative injury in Cr-stressed plants. The destructive effect of ST on oxidative damage was eliminated by the supplementation of SNP plus 5-ALA + ST under Cr stress, possibly by the increased NO accumulation in plants. Exogenously supplied NO mitigated the oxidative injury of Cr-treated tall fescue [67] and tomatoes [48]. This evidently displays that NO synthesis is needed for the positive function of 5-ALA in mitigating the oxidative damage induced by Cr toxicity.

Phytochelatins (PCs) are vital defense substances produced from GSH, which chelates toxic metals, thereby protecting cell organelles against the damaging effect of metal toxicities including Cr [68]. After the binding of PCs with Cr, the complex of PC–Cr is transferred to the vacuole, decreasing the cytoplasmic free Cr [69]. GSH can also bind Cr with the thiol (-SH) group [66], thus, GSH, a precursor of PCs, is able to bind Cr and move it to the vacuole in a non-toxic form [70]. Chromium detoxification might be due to the synthesis of PCs boosted by Cr toxicity [71]. Our results clearly indicate that PCs and GSH jointly function to detoxify Cr in tomato seedlings. Chromium-treated plants showed an increase in GSH, which is being typically changed to GSSG for the detoxification of Cr. This is a likely reason for elevated GSSG levels in Cr-stressed tomato seedlings compared to control plants. Higher GSSG content and lower GSH/GSSG ratio are considered as markers of Cr-induced damage in plants [56]. The pre-treatment of 5-ALA inverted the ratio of GSH/GSSG and the GSH content by raising the GSH/GSSG ratio and GSH content. Analogous data have been revealed by Wu et al. [72], wherein 5-ALA improved GSH in salt-stressed cucumber plants. Cr-stressed tomato seedlings showed higher PCs, and those treated with 5-ALA showed even higher PC levels over the control plants, thereby leading to Cr chelation in a nontoxic form. Similarly, El-Amier et al. [17] documented that the treatment of 5-ALA enhanced Cd and Ni chelation in *Pisum sativum*. 5-ALA together with SNP boosted more PC synthesis, documenting the joint effect of 5-ALA and SNP on PC synthesis under Cr stress. However, ST pre-treatment eliminated the protective role of 5-ALA-induced GSH and PC synthesis by blocking the NR activity and NO synthesis in Cr-stressed seedlings. The supplementation of SNP plus 5-ALA + ST inverted the negative impact of ST on PC and GSH synthesis under Cr stress. The supplementation of SNP that boosted PC and GSH synthesis has been earlier documented in plants treated with Cr [66].

Chromium-stressed tomato plants exhibited differences in the activities of the AsA-GSH cycle-connected enzymes, APX, GR, MDHAR, and DHAR, while analogous data have been reported for maize [73] and rice [50]. Both GR and APX enzymes play a crucial function in eliminating ROS in the AsA-GSH cycle [74]. Tomato plants treated with Cr showed elevated activities of both these enzymes, as has also been shown in *Brassica napus* [75]. The APX and GR activities in cucumbers increased with 5-ALA under low-temperature stress [76] and in tomatoes treated with NO under Cr stress [66]. The supplementation of 5-ALA in Cr-treated plants could not increase the enzymatic antioxidant activities to the level needed to eliminate the destructive impact of the oxidative damage and to retain H_2_O_2_ (ROS) to the level of control seedlings (Figure 4A), as was documented by the increased MDA content (Figure 4B). However, the role of enzymatic antioxidants is not to totally remove ROS, but to achieve an appropriate equilibrium between formation and removal so as to balance the process of photosynthesis, allowing an effective spreading of signals to the nucleus [29,77,78,79]. Nowadays, it is broadly recognized that ROS do not only generate oxidative stress [30,80,81,82], which is boosted under heavy metal toxicity [83,84], but they are necessary for optimal plant growth and development [80,85,86]. Consequently, a low ROS level is needed for optimal plant growth, while a small intensified ROS level is helpful for activating stress defense responses [78,85,87]. Among ROS, H_2_O_2_ is the most stable with the longest lifetime and can easily diffuse through membranes [30,80,88,89], and has been frequently observed to diffuse through leaf veins, acting as a molecule that triggers a long-distance stress defense response [87,90,91] or induces programmed cell death in plants [79,92].

Another two crucial enzymes associated with the AsA-GSH cycle are DHAR and MDHAR. Dehydroascorbate reductase (DHAR) is involved in the recycling of ascorbate (AsA), regenerating a pool of reduced AsA, and detoxifying ROS. DHAR alters DHA to AsA, and MDHAR plays a crucial role in sustaining the ascorbate redox status and reduced pool of AsA [93]. Chromium-stressed tomato plants displayed an elevation in the activities of MDHAR and DHAR, as has already been documented in tomatoes [66]. In tomato plants sprayed with 5-ALA or 5-ALA + SNP under Cr stress, a further increase in DHAR and MDHAR activities was observed. The effect of 5-ALA on MDHAR and DHAR activity has been reported in various crops under various stresses, including tomatoes under low-temperature stress [94], cucumbers under salinity stress [72], and peppers under chilling stress [95]. Chromium’s enhancement of oxidative stress markers in tomatoes is due to down-regulation in the AsA-GSH cycle [11].

Chromium reduced ASA and increased DHA synthesis in tomato plants, as also reported in choysum (*Brassica parachinensis*) plants [96]. Despite this, the supplementation of 5-ALA along with Cr reduced the DHA level and elevated the AsA level by increasing DHAR and MDHAR activities, raising the AsA/DHA ratio. Similar results were reported in salt-stressed cucumber plants by Wu et al. [72]. The pre-treatment with ST along with 5-ALA eliminated the AsA-GSH cycle-associated enzymes, suggesting that the NR-induced NO synthesis is needed for enhancing Cr tolerance in tomatoes with 5-ALA. The enhancement of NR activity by 5-ALA has already been recorded in salt-stressed watermelons [61]. External supply of NO up-regulated the activities of the ASA-GSH cycle-linked enzymes in maize [49]. Exogenous application of 5-ALA significantly enhanced photosynthetic parameters, plant growth, and biomass, while it decreased Cr-induced oxidative stress in maize by improving the antioxidant enzymes’ activities [23].

Environmental stresses such as drought stress and heavy metals are known to reduce the maximum efficiency of PSII photochemistry (F*v*/F*m*) [97,98,99,100]. Chromium treatment significantly decreased F*v*/F*m* (Figure 2F), however, the application of 5-ALA in Cr-treated tomato seedlings mitigated the F*v*/F*m* reduction (Figure 2F). Similarly, it has been shown recently that application of 5-ALA enhanced PSII photochemical activity and delayed the senescence of *Pseudostellaria heterophylla* [101].

We may conclude from our experimental data that the mechanism of 5-ALA-induced Cr tolerance to crop plants is based on the enhancement of NR-generated NO.

## 4. Materials and Methods

### 4.1. Plant Cultivation and Treatments

The experiments were conducted with tomato seedlings (*Solanum lycopersicum* L. cv “SC 2121”) in a growth chamber with a 16/8 h day/night photoperiod, 24 ± 1/16 ± 1 °C day/night temperature, 65 ± 5/75 ± 5% day/night relative humidity, and photosynthetic photon flux density (PPFD) 300 ± 20 μmol photons m^−2^ s^−1^. The seeds were surface-sterilized with 1% NaOCl prior to germination. The tomato seeds were germinated on glass beads soaked in 10% Hoagland’s nutrient solution for a week. Then, the seedlings were transferred into 10 L pots for another three days and later, when they were 10 days old, they were relocated to black polyethylene pots (5 L capacity) with 1/2 strength Hoagland’s nutrient solution [102]. The pH of the nutrient solution was adjusted to 5.5 using 0.01 M KOH. A completely random block trial was designed with three replications, and each replication contained 3 pots.

Ten-day-old tomato seedlings were sprayed once a day for 3 days with 5-aminolevulinic acid hydrochloride (20 mg L^−1^ 5-ALA). The solution of 5-ALA (Sigma) was prepared in 0.01% Tween-20 as a surfactant. An equal amount of surfactant solution was sprayed onto the control plants. Prior to transportation of the 10-day-old seedlings, the roots were kept in the NR inhibitor sodium tungstate (0.1 mM ST) for 3 h, as described before [64]. Thirteen-day-old tomato seedlings were treated with or without chromium Cr(VI) (50 µM K_2_Cr_2_O_7_) for two additional weeks. The nutrient solution was changed every 3 days with a newly prepared solution. Tomato seedlings were sprayed with 0.1 mM sodium nitroprusside (SNP) solution once a week for two weeks after transplantation of the seedlings.

Three plants per treatment (one from each replicate) were used to estimate dry weight. The samples were air-dried and subjected to 105 °C in an oven for 10 min, and then an additional 72 h at 72 °C for the estimation of the dry weights. The remaining two seedlings from each replicate (6 seedlings total) in each treatment were used for the other measurements.

A brief description of the treatments is illustrated in Figure 8.

### 4.2. Estimation of Chlorophyll Content and Chlorophyll Fluorescence

The leaf material (1 g) was homogenized in an acetone (90%) solution. After filtrating the extract, the absorbances were noted on a spectrophotometer (Shimadzu UV-Visible, Japan) to estimate the content of leaf chlorophyll, employing the protocol detailed by Strain and Svec [103].

The maximum efficiency of PSII photochemistry (F*v*/F*m*) was estimated in dark adapted (30 min) leaves using a Mini-PAM instrument (Heinz Walz GmbH, Forchheim, Germany).

### 4.3. Estimation of Relative Water Content (RWC) and Leaf Water Potential

The method of Barrs and Weatherley [104] was adopted to measure RWC. Leaves was collected from the same position on the plant and weighed instantly to estimate the leaf fresh weight (FW). Thereafter, the same leaf materials were kept in bottles containing distilled water for a day. Afterwards, the leaf surface water was removed, and samples were reweighed to record the leaf turgid weight (TW). Lastly, the leaf dry weight (DW) was estimated after drying out the leaf samples to a constant weight at 65 °C. The equation below was used to compute RWC:RWC = [(FW − DW)/(TW − DW)] × 100 (1)

A pressure chamber instrument (PMS model 600, PMS Instrument Company, Albany, OR, USA) was used to estimate leaf water potential on the 3rd leaf from the top of each plant at 8.00 a.m.

### 4.4. Estimation of Leaf Proline

Leaf proline content was quantified following the method of Bates et al. [105]. Fresh leaf tissue (500 mg) was homogenized in 10 mL of sulfosalicylic acid (3%). The filtrate (2 mL) was added into equal amounts of acid-ninhydrin and glacial acetic acid and, subsequently, the sample solution was incubated at 100 °C for one hour. Following cooling of the sample solution, 4 mL toluene was added. The absorbance was read at 520 nm using a spectrophotometer.

### 4.5. Estimation of NO Content

Leaf NO was estimated following the modified procedure of Zhou et al. [106]. Briefly, fresh leaf tissue (600 mg) was extracted in 4% zinc diacetate and 3 mL of 50 mM acetic acid buffer (pH 3.6). The extract solution was centrifuged at 10,000× *g* for 15 min. After washing the pellet with the extraction buffer (1.0 mL), 100 mg of charcoal was treated with the extract after both aliquots were mixed. After filtration followed by vortexing, the Greiss reagent and the mixture (1.0 mL) were added and kept at room temperature for 30 min. Absorbances were recorded at 540 nm.

### 4.6. Quantification of Nitrate Reductase (NR)

Sampling for nitrate reductase (NR) activity was conducted 6 h after induction of the light period as described by Sun et al. [107]. In brief, an extraction buffer consisting of EDTA (1 mM), 1,4-dithiothreitol (DTT; 5 mM), glycerol (10%), 0.1 M HEPES–KOH (pH 7.5), phenylmethylsulfonyl fluoride (0.5 mM), Triton X-100 (0.1%, *v*/*v*), flavin adenine dinucleotide (FAD; 20 μM), and polyvinylpyrrolidone (PVP; 1%) was used. After appropriate extraction, the extract was centrifugated at 12,000× *g* at 4 °C for 20 min. An aliquot was used to analyze activity of NR at 520 nm.

### 4.7. Quantification of Chromium Distribution

The subcellular fractionation of Cr in leaf and root tissues was quantified employing the procedure of Sheng et al. [108]. Briefly, fresh leaf and root material (0.5 g) was extracted in 10 mL of extraction buffer (1.0 mM DL-dithioerythritol, 0.25 mM sucrose, 5 mM AsA, and 50 mM Tris-HCl). The extract was filtered by using a nylon fabric with a mesh size of 100 mm and the filtrate was labelled as “cell wall fraction” (CWF) and subjected to centrifugation (15,000 rpm) for 45 min. The aliquot solution and resultant pellet were labelled as “soluble fraction” (SF) and “organelle fraction” (OF), respectively. These procedures were carried out at 4 °C. Both CWF and OF were transferred to an Erlenmeyer flask (100 mL) filled with deionized water, dried, and digested with HNO_3_ (5 mL) [109]. Thereafter, Cr concentrations were determined with an ICP-OES (Perkin Elmer Optima 5300 DV).

### 4.8. Quantification of Leaf H_2_O_2_ and MDA

The content of leaf H_2_O_2_ was quantified adopting the protocol descripted by Velikova et al. [110]. A sample of 500 mg of fresh leaf material was homogenized in 1% TCA (3 mL). After centrifugation, 0.75 mL of the filtrate was treated with 1.5 mL of 1 M KI and 10 mM K buffer (0.75 mL), and then the absorbances were read at 390 nm.

The method outlined by Weisany et al. [111] was adopted for the estimation of leaf MDA.

### 4.9. Electrolyte Leakage (EL)

The EL was estimated employing the method of Dionisio-Sese and Tobita [112]. Deionized water was used for leaf tissue surface cleaning. Leaf discs were kept in bottles, each filled with 10 mL of deionized water, on a rotary shaker at room temperature for 24 h to estimate the electrolyte conductivity 1 (EC1). Afterwards, these samples were kept in an autoclave for 20 min at 120 °C to estimate the last electrolyte conductivity (EC2). The EL was computed using the below formula:EL (%) = (EC1/EC2) × 100

### 4.10. Determination of 5-ALA

An amount of 100 mg fresh leaf tissue was extracted in sodium acetate buffer (2 mL, 1*M* and pH 4.6) prior to centrifugation at 12,000× *g* for 10 min. The mixture consisted of aliquot (100 µL), acetylacetone (25 μL), and distilled water (400 µL). Thereafter, this mixture solution was heated to boiling in a hot water bath for 10 min. This mixture was cooled at room temperature and then an equivalent amount of modified Ehrlich’s reagent was added and vortexed for 2 min. Afterwards, the treated solution was incubated for 10 min, its absorbance was read at 555 nm, and 5-ALA content was calculated from a standard curve of 5-ALA [113].

### 4.11. Determination of Ascorbic Acid and Glutathione

Meta-phosphoric acid buffer (3 mL, 5%) and 1 mM EDTA were used to homogenize 500 mg of fresh leaf. The homogenized mixture was subjected to a centrifuge at 11,500× *g* at 4 °C for 12 min. The resulting reaction mixture was used to quantify glutathione and ascorbate.

Potassium-phosphate buffer (pH 7.0; 500 mM) was used to quantify ascorbate following Huang et al. [114]. The assay of reduced ascorbate was conducted in ascorbate oxidase (0.5 units) and potassium-phosphate buffer (pH 7.0; 0.1 M). The treated samples were read at 265 nm.

To estimate total AsA, the samples were extracted with 30 mM dithiothreitol. Dehydroascorbate (DHA) was computed by deducting reduced AsA content from total AsA.

The method of Yu et al. [115] was followed for assaying reduced GSH and glutathione disulfide (GSSG). A K-phosphate buffer (0.6 mL, 0.5 M, pH 7.0) was added to 0.4 mL of the sample extract. The GSH was measured by the changes in OD values at 412 nm for NTB (2-nitro-5-thiobenzoic acid) generated by the Ellman’s reagent (5,5-dithio-bis-(2-nitrobenzoic acid, DTNB) reduction. The GSSG level was computed by subtracting the GSH concentration from that of the derivatizing agent 2-vinylpyridine.

### 4.12. Extraction of Leaf Samples and Enzyme Assays

The fresh leaf material was extracted in a buffer containing glycerol (10%), KCl (100 mM), potassium phosphate buffer (50 mM, pH 7.0), β-mercaptoethanol (5 mM), and ascorbic acid (1 mM). After the extract was properly centrifugated, the protein content was estimated by using aliquots, as described by Bradford [116].

The activity of the ascorbate peroxidase (APX) enzyme was estimated by adopting the method of Nakano and Asada [117]. The mixture solution consisted of EDTA (0.1 mM), K–P buffer (50 mM, pH 7.0), H_2_O_2_ (0.1 mM), and ascorbic acid (0.5 mM). Absorbance was recorded at 290 nm for 1 min.

The estimation of glutathione reductase (GR) activity was calculated by following the protocol of Hasanuzzaman et al. [118]. The extract was added to a mixer solution of 100 mM, K–P buffer at pH 7, 200 µM NADPH, 1.0 mM EDTA, and 100 µM GSSG. Alterations in absorbance were recorded at 340 nm.

The estimation of monodehydroascorbate reductase (MDHAR) was performed by employing the protocol outlined by Hossain et al. [119]. The solution of the assay consisted of NADPH (0.2 mM), 50 mM Tris–HCl buffer (pH 7.5), AsA (2.5 mM), 0.5 units of AO, and the enzyme extract. H_2_O_2_ was added to the assay solution to initiate the reaction. Thereafter, the absorbances were read at 340 nm for 1 min.

The activity of dehydroascorbate reductase (DHAR) was quantified by employing the protocol outlined by Nakano and Asada [117]. The extract was added to a mixture solution consisting of 2.5 mM GSH, K–P buffer (50 mM), and 0.1 mM DHA at pH 7.0. Thereafter, the absorbance was read at 265 nm.

### 4.13. Statistical Treatment of the Data

All data were checked for normality and a two-way ANOVA was performed with treatment and stress as factors. A post hoc analysis was performed to determine statistical significance with Tukey’s test. Significance was estimated at a level of *p* < 0.05. Standard errors and means were used to present the data. Data analysis and graphs were obtained using SAS version 9.1.

## 5. Conclusions

5-ALA, by enhancing NR activity and NO synthesis in tomato plants, improved tolerance to Cr stress. In addition, 5-ALA increased GSH and PCs and, thus, mediated Cr sequestration and contributed to its detoxification. The supplementation of ST inhibited NR activity and NO synthesis and eliminated the positive impact of 5-ALA on Cr-stressed plants. 5-ALA-induced NR-driven NO generation can successfully modulate the AsA-GSH cycle to reduce oxidative damage and confer Cr tolerance. Thus, NR and NO are twin players, reducing Cr toxicity in tomato plants via antioxidant signalling cascades. However, more experiments and field tests must be implemented to formulate 5-ALA use in agriculture, in order to obtain sustainable crop production in Cr-contaminated soils. In addition, the role of other crucial enzymes involved in 5-ALA-induced plant tolerance to Cr toxicity should be investigated.

## Figures and Tables

**Figure 1 plants-12-00502-f001:**
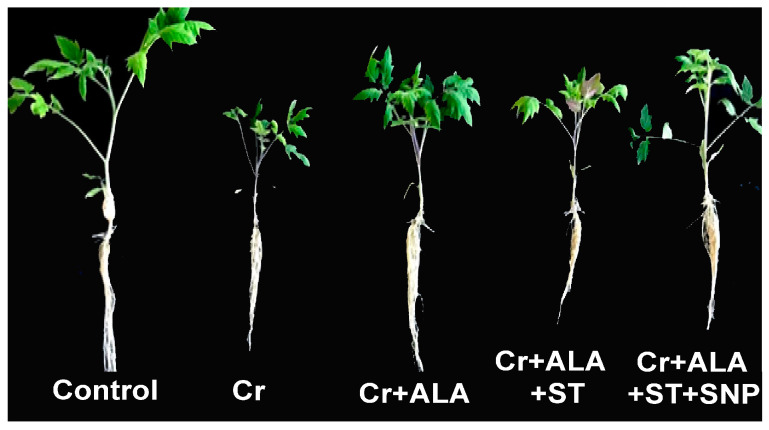
The effects of chromium (Cr) alone, or combined with 5-aminolevulinic acid (5-ALA), sodium tungstate (ST), and sodium nitroprusside (SNP) on tomato seedlings. Chromium was applied for 14 days on 13-day-old tomato seedlings pre-treated with 5-ALA for 3 days. Photographs were taken at the end of the experiments with both control seedlings and treatments being 27 days old.

**Figure 2 plants-12-00502-f002:**
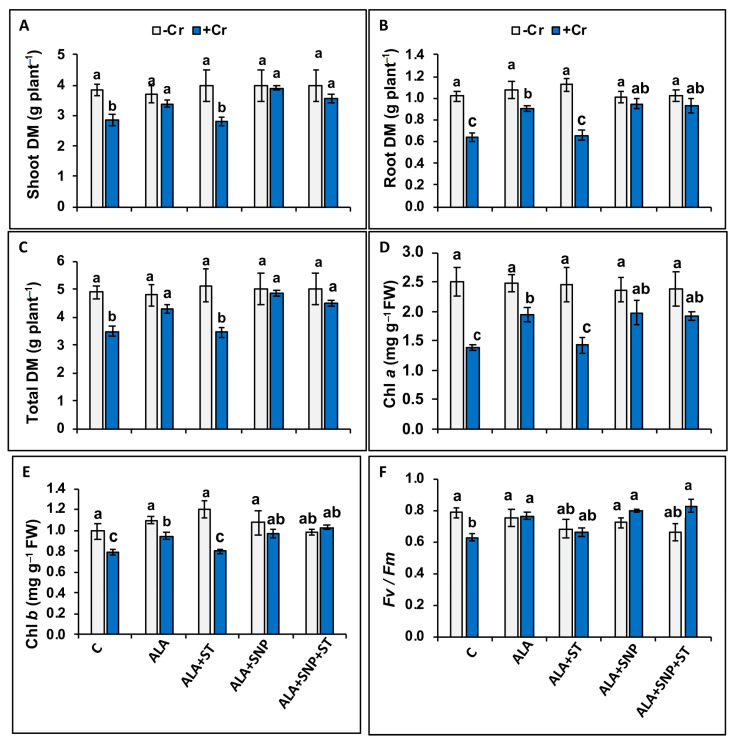
Shoot (**A**), root (**B**), total dry mass (DM) (**C**), chlorophyll *a* (**D**), chlorophyll *b* (**E**), and maximum efficiency of photosystem II photochemistry (F*v*/F*m*) (**F**) in Cr-treated (+Cr) and non-treated (−Cr) tomato seedlings, sprayed with 20 mg L^−1^ 5-aminolevulinic acid (5-ALA) or 0.1 mM sodium tungstate (ST) + 5-ALA, 5-ALA + 0.1 mM sodium nitroprusside (SNP), or 5-ALA + SNP + ST. Data are means ± S.E, different lower-case letters on bars reflect that mean values differ significantly at *p* ≤ 0.05.

**Figure 3 plants-12-00502-f003:**
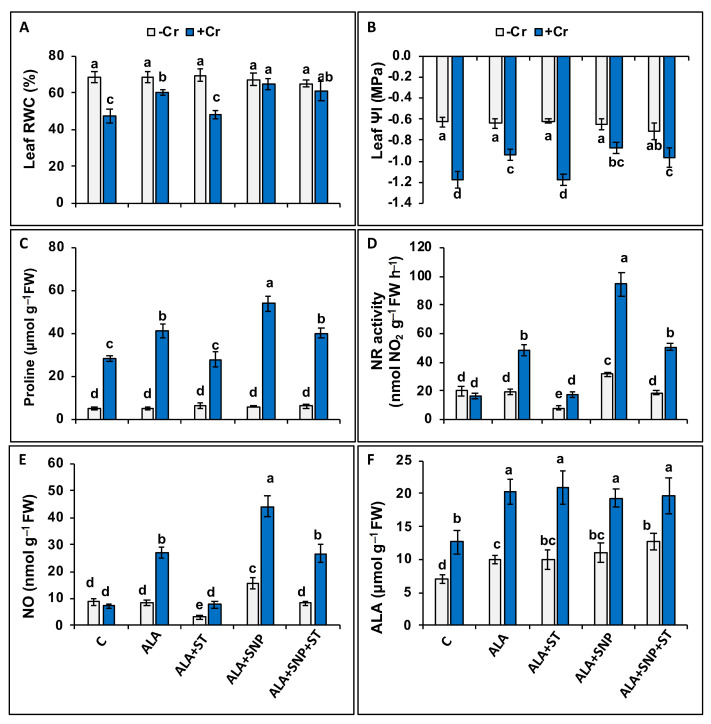
Leaf relative water content (RWC) (**A**), leaf water potential (Leaf Ψl) (**B**), free proline (**C**), nitrate reductase (NR) activity (**D**), nitric oxide (NO) content (**E**), and 5-aminolevulinic acid (5-ALA) content (**F**) in Cr-treated (+Cr), and non-treated (−Cr) tomato seedlings, sprayed with 20 mg L^−1^ 5-aminolevulinic acid (5-ALA), 0.1 mM sodium tungstate (ST) + 5-ALA, 5-ALA + 0.1 mM sodium nitroprusside (SNP), or 5-ALA + SNP + ST. Data are means ± S.E, different lower-case letters on bars reflect that mean values differ significantly at *p* ≤ 0.05.

**Figure 4 plants-12-00502-f004:**
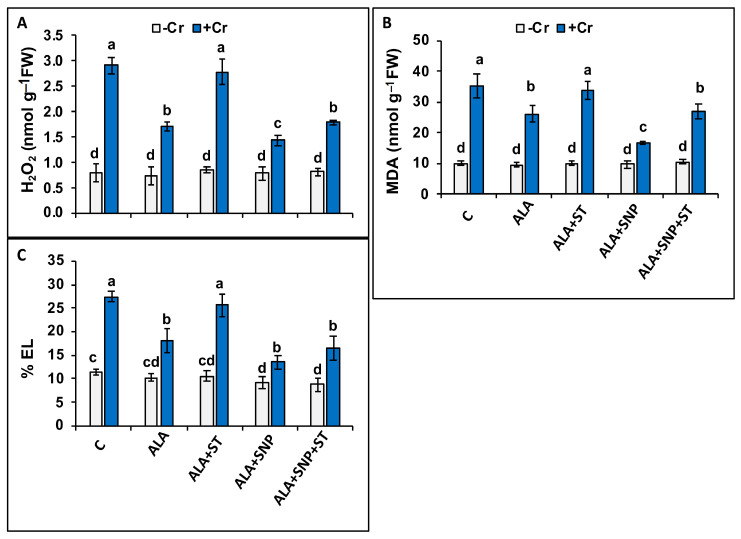
Hydrogen peroxide (H_2_O_2_) (**A**), malondialdehyde (MDA) (**B**), and electrolyte leakage (EL) (**C**) in Cr-treated (+Cr), non-treated (−Cr) tomato seedlings, sprayed with 20 mg L^−1^ 5-aminolevulinic acid (5-ALA), 0.1 mM sodium tungstate (ST) + 5-ALA, 5-ALA + 0.1 mM sodium nitroprusside (SNP), or 5-ALA + SNP + ST. Data are means ± S.E, different lower-case letters on bars reflect that mean values differ significantly at *p* ≤ 0.05.

**Figure 5 plants-12-00502-f005:**
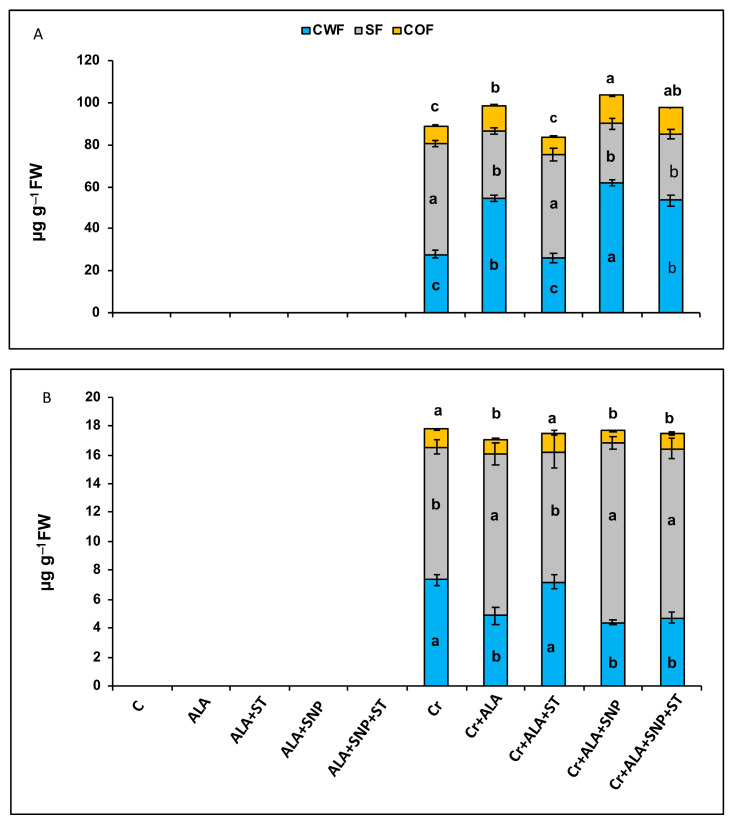
The cell wall fraction (CWF), cell organelle fraction (COF), and vacuolar soluble fraction (SF) of chromium (Cr) in roots (**A**) and leaves (**B**) in Cr-treated (+Cr) and non-treated (−Cr) tomato seedlings sprayed with 20 mg L^−1^ 5-aminolevulinic acid (5-ALA), 0.1 mM sodium tungstate (ST) + 5-ALA, 5-ALA + 0.1 mM sodium nitroprusside (SNP), or 5-ALA + SNP + ST. Data are means ± S.E, different lower-case letters on bars reflect that mean values differ significantly at *p* ≤ 0.05.

**Figure 6 plants-12-00502-f006:**
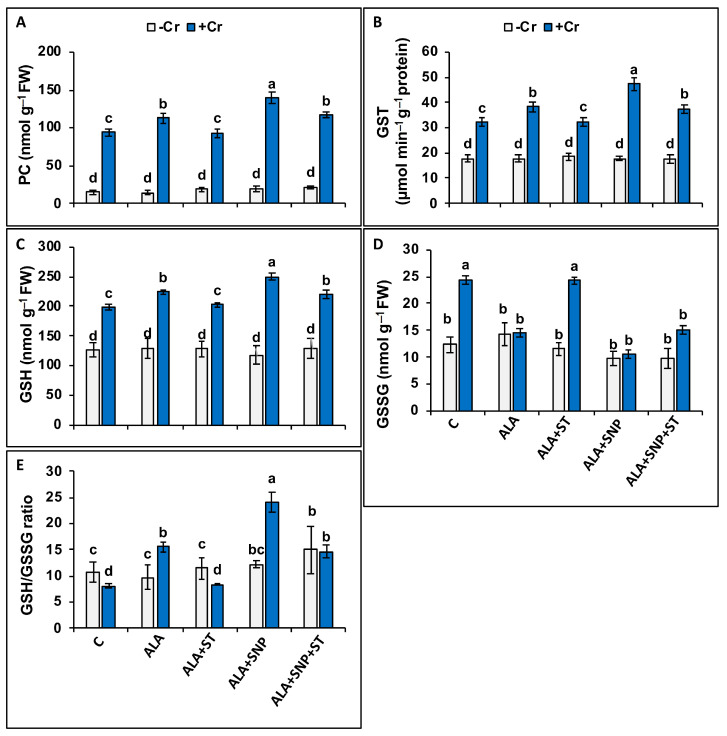
Phytochelatins (PC) (**A**), glutathione-*S*-transferase (GST) (**B**), reduced glutathione (GSH) (**C**), oxidized glutathione (GSSG) (**D**), and GSH/GSSG (**E**) in Cr-treated (+Cr) and non-treated (−Cr) tomato seedlings sprayed with 20 mg L^−1^ 5-aminolevulinic acid (5-ALA), 0.1 mM sodium tungstate (ST) + 5-ALA, 5-ALA + 0.1 mM sodium nitroprusside (SNP), or 5-ALA + SNP + ST. Data are means ± S.E, different lower-case letters on bars reflect that mean values differ significantly at *p* ≤ 0.05.

**Figure 7 plants-12-00502-f007:**
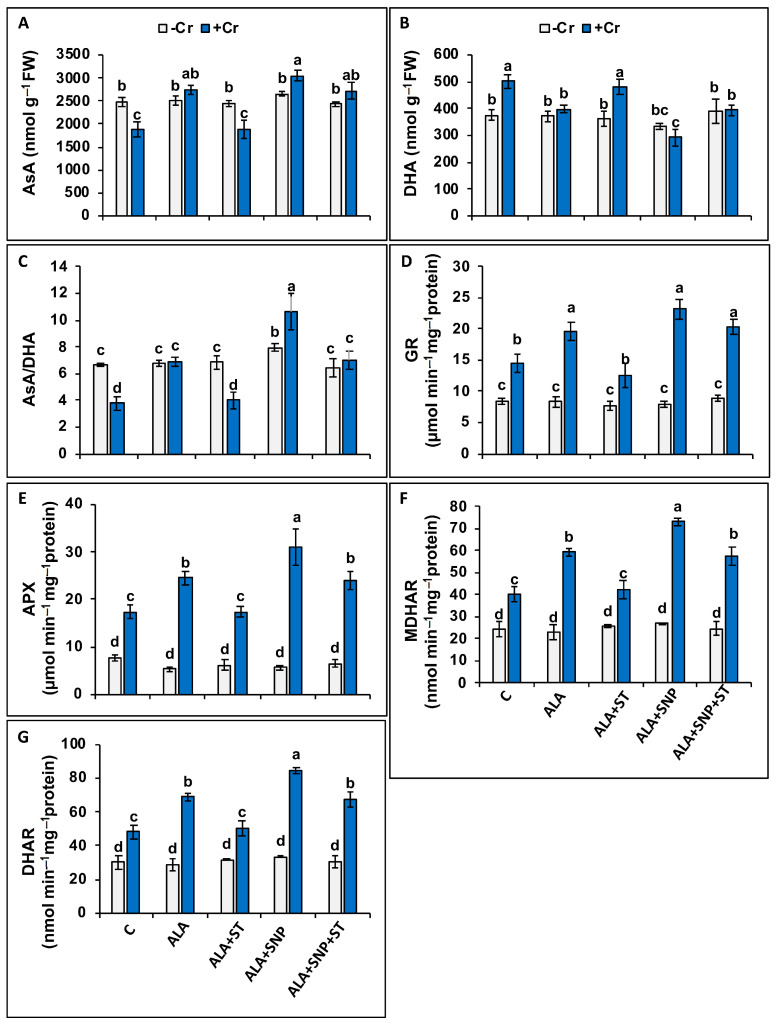
Ascorbate (AsA) (**A**), dehydroascorbate (DHA) (**B**), AsA/DHA ratio (**C**), activities of glutathione reductase (GR) (**D**), ascorbate peroxidase (APX) (**E**), monodehydroascorbate reductase (MDHAR) (**F**), and dehydroascorbate reductase (DHAR) (**G**) in Cr-treated (+Cr) and non-treated (−Cr) tomato seedlings sprayed with 20 mg L^−1^ 5-aminolevulinic acid (5-ALA), 0.1 mM sodium tungstate (ST) + 5-ALA, 5-ALA + 0.1 mM sodium nitroprusside (SNP), or 5-ALA + SNP + ST. Data are means ± S.E, different lower-case letters on bars reflect that mean values differ significantly at *p* ≤ 0.05.

**Figure 8 plants-12-00502-f008:**
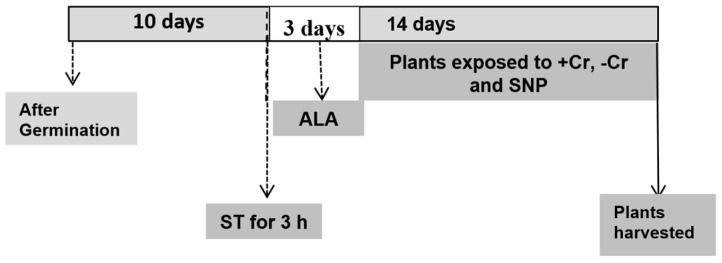
An outline of the experimental design showing that 10-day-old tomato seedlings were treated for 3 h with 0.1 mM sodium tungstate (ST) before the application of 20 mg L^−1^ 5-aminolevulinic acid (5-ALA) for 3 days. Different chemicals were applied to examine the effect of 5-ALA, with or without chromium (Cr) treatment, on tomato seedlings. The concentrations used were 20 mg L^−1^ 5-aminolevulinic acid (5-ALA), 50 µM Cr(VI), 0.1 mM sodium nitroprusside (SNP, a donor of NO), and 0.1 mM sodium tungstate (ST, an inhibitor of nitrate reductase; NR, an enzyme that catalyzes NO formation). Chromium (Cr) was applied for 14 days on 13-day-old tomato seedlings. At the end of the experiment, tomato seedlings were 27 days old.

## Data Availability

The data supporting reported results can be found in the article.

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
