# Peer review of "5-Aminolevulinic Acid Induces Chromium [Cr(VI)] Tolerance in Tomatoes by Alleviating Oxidative Damage and Protecting Photosystem II: A Mechanistic Approach"

_plants, 2023, doi:10.3390/plants12030502_

Round 1
Reviewer 1 Report (Previous Reviewer 3)
The work has been performed for understanding the role of 5-ALA-in tomato seedlings exposed to chromium stress conditions. Sodium tungstate and sodium nitroprusside in the effect on the regulation of crop response to 5-ALA. The manuscript needs some improvement in particular:
-the introduction needs to be improved on the physiological and biochemical role of 5-aminolevulinic acid in normal or stress conditions;
-Improve the hypothesis of the work highlighting the possible role of 5-ALA;
Materials and methods
The development stage of the seedling is not optimal since the leaves are not fully expanded and with the reduced functionality as reported by the lower FV/FM ratio, normally it should be above 0.83.
-Please report the interaction values in the legends or in the graph of Stress vs treatment. The authors reported that the normality was checked but with so low number of data I assume that the distribution was not confirmed, so please demonstrated it or remove the statement.
Results are satisfactory and differences were reported as percentage.
Discussion please try to explain the biological role of 5-ALA and its effect. Please avoid to misleading the results, for example at the end of discussion 5-ALA did not increase the Fv/FM ratio but preserved the values or avoided the decline.
Author Response
The work has been performed for understanding the role of 5-ALA-in tomato seedlings exposed to chromium stress conditions. Sodium tungstate and sodium nitroprusside in the effect on the regulation of crop response to 5-ALA. The manuscript needs some improvement in particular:
-the introduction needs to be improved on the physiological and biochemical role of 5-aminolevulinic acid in normal or stress conditions
Introduction was improved on the physiological and biochemical role of 5-aminolevulinic acid under stressed conditions (lines 66-77).
-Improve the hypothesis of the work highlighting the possible role of 5-ALA.
Working hypothesis was improved.
Materials and methods
The development stage of the seedling is not optimal since the leaves are not fully expanded and with the reduced functionality as reported by the lower FV/FM ratio, normally it should be above 0.83.
We have added on lines 147-150: “The Fv/Fm value of control leaves of tomato seedlings sprayed with the surfactant solution (0.01% Tween-20) was 0,799 ± 0.009 (Figure 2F), that is 4% lower than those that were not treated with Tween-20 (0.831 ± 0.003).
Thus, the Fv/Fm value of control leaves not treated with Tween-20 was above 0.83.
However, according to:
- Krause, G.H.; Weis, E. Chlorophyll fluorescence as a tool in plant physiology. II. Interpretation of fluorescence signals. Photosynthesis Research 1984, 5, 139-157.
- Krause G.H.; Weiss E. Chlorophyll fluorescence and photosynthesis: the basics. Annual Review of Plant Physiology and Plant Molecular Biology1991, 42, 313–349.
Fv/Fm ratio at room temperature is about 0.8
3 . Garmash, E.V.; Dymova, O.V.; Silina, E.V.; Malyshev, R.V.; Belykh, E.S.; Shelyakin, M.A.; Velegzhaninov, I.O. AOX1a expression in Arabidopsis thaliana affects the state of chloroplast photoprotective systems under moderately high light conditions. Plants 2022, 11, 3030.
"In all the WT samples, the Fv/Fm reached 0.8, which is the typical value for a healthy mature leaf."
-Please report the interaction values in the legends or in the graph of Stress vs treatment. The authors reported that the normality was checked but with so low number of data I assume that the distribution was not confirmed, so please demonstrated it or remove the statement.
We performed post-hock tests only after we confirmed a significant effect between stress vs treatment for each individual dependent variable.
Results are satisfactory and differences were reported as percentage.
Discussion please try to explain the biological role of 5-ALA and its effect. Please avoid to misleading the results, for example at the end of discussion 5-ALA did not increase the Fv/FM ratio but preserved the values or avoided the decline.
Text on the biological role of 5-ALA and its effect was added in both Introduction and Discussion sections. Discussion was checked for misleading results and the interpretation of the effect of 5-ALA on the Fv/FM ratio was corrected.
Reviewer 2 Report (Previous Reviewer 2)
The manuscript entitled “5-Aminolevulinic acid induces chromium tolerance in tomato by alleviating oxidative damage and protecting photosystem II: A mechanistic approach” presents a complex investigation on the potential application of 5-ALA in reduction of Cr(VI) toxicity. The authors investigated the mechanism by using sodium nitroprusside as a NO donor or/and sodium tungstate as a NR inhibitor. Generally, the manuscript is well prepared, however a few significant drawbacks can be found that need correction before resubmitting the manuscript.
11. The first one is to indicate in the title and abstract the valency of applied chromium.
22. The authors confuse Cr(VI) application/spiking and stress (the action and its effect). It should be carefully reedited through the manuscript. E.g. L199 – “chromium stress plants” should be changed into “chromium treated plants”
33. Delete numerous sentences in all manuscript section such as e.g. “seems to be the first” - L30, “should have been the effect” – L139…
44. L37 – anthropogenic effect are not the cause of Cr toxicity
55. L42 – absorption concerns light/energy, plants can uptake elements from soil
66. The introduction is in many parts repeatable, one sentence repeats the half of the previous one. It’s too long in general and consists many “basic” knowledge”. The same for Discussion section – it should be shortened.
77. L80-83 should be moved to upper paragraph.
88. Change “tomatoes” to tomato seedlings
99. Not every picture of seedlings from all experimental variants was shown in Figure 1.
110. Please check the post-hock test results in figures (Fig. 2E,F, 3E, 4C, 6D).
111. In the Results section please unify the sense of the captions, should it be part of the investigations or a conclusion (e.g. 2.9)? In my opinion the first one.
112. The first caption in the Discussion section is a repetition of the introduction. Should be removed.
113. L282 – Cd or Cr?
114. Remove from the text sentences such as “according to our knowledge..”, “there seems to be no…”, “it appears to be..”. This should not occur in scientific publication.
Author Response
The manuscript entitled “5-Aminolevulinic acid induces chromium tolerance in tomato by alleviating oxidative damage and protecting photosystem II: A mechanistic approach” presents a complex investigation on the potential application of 5-ALA in reduction of Cr(VI) toxicity. The authors investigated the mechanism by using sodium nitroprusside as a NO donor or/and sodium tungstate as a NR inhibitor. Generally, the manuscript is well prepared, however a few significant drawbacks can be found that need correction before resubmitting the manuscript.
Thank you for your constructive comments that improved our manuscript.
- The first one is to indicate in the title and abstract the valency of applied chromium.
The valency of applied chromium has been indicated in the title and the abstract.
- The authors confuse Cr(VI) application/spiking and stress (the action and its effect). It should be carefully reedited through the manuscript. E.g. L199 – “chromium stress plants” should be changed into “chromium treated plants”
It was changed to “chromium treated plants” and the whole manuscript was checked.
- Delete numerous sentences in all manuscript section such as e.g. “seems to be the first” - L30, “should have been the effect” – L139…
Both sentences were changed and marked in yellow.
- L37 – anthropogenic effect are not the cause of Cr toxicity.
The sentence on line 37 was changed.
- L42 – absorption concerns light/energy, plants can uptake elements from soil
The sentence was changed.
- The introduction is in many parts repeatable; one sentence repeats the half of the previous one. It’s too long in general and consists many “basic” knowledge”. The same for Discussion section – it should be shortened.
The repeatable parts in Introduction and Discussion section were deleted but, in both sections, new text was added as requested by the other reviewers.
- L80-83 should be moved to upper paragraph.
The two sentences were moved to the previous paragraph.
- Change “tomatoes” to tomato seedlings
The word “tomatoes” was changed to “tomato seedlings”.
- Not every picture of seedlings from all experimental variants was shown in Figure 1.
In Figure 1 are shown the effects of Cr treatments only, while is missing the effect of Cr+5-ALA+SNP.
- Please check the post-hock test results in figures (Fig. 2E,F, 3E, 4C, 6D).
The post-hock test results were checked and some corrections were made. Thank you for pointing it.
- In the Results section please unify the sense of the captions, should it be part of the investigations or a conclusion (e.g. 2.9)? In my opinion the first one.
Subheadings in the results section were changed according to your suggestion.
- The first caption in the Discussion section is a repetition of the introduction. Should be removed.
The first paragraph in Discussion section was removed together with other repetitions in Discussion.
- L282 – Cd or Cr?
Yes, it should be Cr. Thank you for pointing it.
- Remove from the text sentences such as “according to our knowledge..”, “there seems to be no…”, “it appears to be.”. This should not occur in scientific publication.
We removed such phrases.
Reviewer 3 Report (Previous Reviewer 1)
The submitted manuscript has been significantly edited by the authors, especially in the Discussion section. Unfortunately, the authors did not mark the places that were edited. There are some minor comments. The abstract should begin with 1-2 sentences with a description of the problem, and not with the purpose of the study. Reduce the number of keywords.
Author Response
Τhe submitted manuscript has been significantly edited by the authors, especially in the Discussion section. Unfortunately, the authors did not mark the places that were edited. There are some minor comments. The abstract should begin with 1-2 sentences with a description of the problem, and not with the purpose of the study. Reduce the number of keywords.
Thank you for your constructive comments that improved our manuscript.
Now in the revised manuscript we marked in yellow all places we edit.
We inserted in the abstract 2 sentences with the background of the problem. We also deleted a keyword and we have now one less than the upper limit of 10.
Round 2
Reviewer 2 Report (Previous Reviewer 2)
Authors corrected the manuscript according to the comments.
This manuscript is a resubmission of an earlier submission. The following is a list of the peer review reports and author responses from that submission.
Round 1
Reviewer 1 Report
The presented manuscript is devoted to the toxic effect of chromium ions on tomatoes and the use of ALA to reduce this effect. many parameters were used in the work, there is no doubt about the results obtained, some of them were obtained for the first time, the conclusions correspond to the results. The manuscript is well-formed, there are minor remarks. In the annotation, it is necessary to indicate the prospect of using the results obtained. Many biochemical terms are used in the manuscript, and their full names should be given not only in the Materials and Methods section, but also in the text when they are first used. e.g. Cr-S, MDA, EL, SNP.
The discussion is long, it is desirable to shorten it. The Discussion section does not use subsections.
Reference 58 -year in bold type
Author Response
The presented manuscript is devoted to the toxic effect of chromium ions on tomatoes and the use of ALA to reduce this effect. many parameters were used in the work, there is no doubt about the results obtained, some of them were obtained for the first time, the conclusions correspond to the results. The manuscript is well-formed, there are minor remarks. In the annotation, it is necessary to indicate the prospect of using the results obtained. Many biochemical terms are used in the manuscript, and their full names should be given not only in the Materials and Methods section, but also in the text when they are first used. e.g., Cr-S, MDA, EL, SNP.
Thank you for your comments that helped us to improve the weak points of our manuscript. Full names were given in the first mention of the abbreviations used in the text when. In Conclusions section the prospect of using the obtained results was added
The discussion is long, it is desirable to shorten it. The Discussion section does not use subsections.
The subsections were deleted and some duplications were eliminated but discussion was not shortened significantly because new paragraphs were added about ROS and antioxidants.
Reference 58 -year in bold type.
Reference 58 is a book chapter and the year must not be in bold.
Reviewer 2 Report
The manuscript presents a huge data set on the alleviation of chromium toxicity and its mechanism in tomato plants. The data are novel however, the manuscript suffers a few drawbacks that should be addressed before publishing.
1) The title is too long and has to be shortened to present the main findings.
2) Improve Abstract and Introduction. In present form they are messy and hardly readable.
3) Provide the aim of the study at the end of the Introduction section.
4) In Figure 5 remove “roots” and “leaves” form axis Y and change into headings
5) In Conclusions describe the application of the obtained results. If they are not applicable due to primary/fundamental nature, provide a chart/scheme showing the evaluated mechanisms of Cr stress alleviation by applied chemicals.
6) Language usage needs deep revision by a native speaker.
Author Response
The manuscript presents a huge data set on the alleviation of chromium toxicity and its mechanism in tomato plants. The data are novel however, the manuscript suffers a few drawbacks that should be addressed before publishing.
Thank you for your comments that helped us to improve the weak points of our manuscript.
- The title is too long and has to be shortened to present the main findings.
The title was shortened
- Improve Abstract and Introduction. In present form they are messy and hardly readable.
Abstract and Introduction were revised.
- Provide the aim of the study at the end of the Introduction section.
The aim of the study was provided at the end of the Introduction section (lines 86-91).
- In Figure 5 remove “roots” and “leaves” form axis Y and change into headings
Figure 5 was changed as you suggested.
- In Conclusions describe the application of the obtained results. If they are not applicable due to primary/fundamental nature, provide a chart/scheme showing the evaluated mechanisms of Cr stress alleviation by applied chemicals.
In Conclusions section we inserted a sentence describing application of the results obtained and future prospect.
- Language usage needs deep revision by a native speaker.
Language corrections were done at the whole manuscript.
Reviewer 3 Report
The manuscript reports a work performed on the tomato seedling (14 days) exposed to Cr stress and different treatments including promoters of NO production or inhibitor of the NR that can influence the NO production. The main concern of this work is related to the early development stages. The stress and treatments should be performed in fully expanded and fully active leaves in term of photosynthesis.
Introduction should be improved and revise the text, the state of art should report the NO function and antioxidant systems.
Rows 60-61, the sentence is not clear “ALA is known as a precursor existing in fungus, bacteria, and amino acids of different plants and animals [15,16] and as a key plant growth stimulator against several metal toxicities” please revise.
Row 70: the function of NO in plants must be described better and expanded reporting the relationship of NO and antioxidant systems in plants.
Row 480. Please correct the “Chlorophyll Conteny” in “Chlorophyll content”.
The statistical analysis must be refined, please use ANOVA and post-test Turkey or Bonferroni. A two-way ANOVA should be performed as two factors (treatments vs Stress).
Results and particular Fv/Fm that represents the maximum quantum efficiency of PSII should be above 0.83, the non-stressed plants have a ratio lower than 0.8 indicating the stress conditions or not fully active leaves.
Nitrate reductase (NR) enzyme is tightly connected with light conditions, therefore the sampling time and the light exposure should be considered in the work.
Even if the work suggest promising positive effect of ALA and SNP, the overall experiment design should be re-considered.
Author Response
The manuscript reports a work performed on the tomato seedling (14 days) exposed to Cr stress and different treatments including promoters of NO production or inhibitor of the NR that can influence the NO production. The main concern of this work is related to the early development stages. The stress and treatments should be performed in fully expanded and fully active leaves in term of photosynthesis.
Thank you for your comments that helped us to improve the weak points of our manuscript. The experiments were performed in one-month seedlings and the leaves that were sampled or measured for photosynthesis were fully expanded and fully active.
Introduction should be improved and revise the text, the state of art should report the NO function and antioxidant systems.
Introduction was revised and recent literature on NO function and antioxidant systems was included (paragraphs beginning on line 72.)
Rows 60-61, the sentence is not clear “ALA is known as a precursor existing in fungus, bacteria, and amino acids of different plants and animals [15,16] and as a key plant growth stimulator against several metal toxicities” please revise.
The sentence was rewritten (lines 63-64).
Row 70: the function of NO in plants must be described better and expanded reporting the relationship of NO and antioxidant systems in plants.
The function of NO and the relationship of NO with antioxidant systems in plants was described better in two consecutive paragraphs beginning on line 72.
Row 480. Please correct the “Chlorophyll Conteny” in “Chlorophyll content”.
Yes, it was corrected.
The statistical analysis must be refined, please use ANOVA and post-test Turkey or Bonferroni. A two-way ANOVA should be performed as two factors (treatments vs Stress).
A two-way ANOVA was performed and the text in statistical analysis subsection was changed accordingly.
Results and particular Fv/Fm that represents the maximum quantum efficiency of PSII should be above 0.83, the non-stressed plants have a ratio lower than 0.8 indicating the stress conditions or not fully active leaves.
Non stressed plants have Fv/Fm values around 0.83. However, our control plants have been sprayed with 0.01% Tween-20, that was used as surfactant in treatments. Before spraying with Tween-20, the Fv/Fm values were around 0.83, but this value was not used as control, because the difference between ALA treated and control plants should have been the effect of ALA alone.
Nitrate reductase (NR) enzyme is tightly connected with light conditions, therefore the sampling time and the light exposure should be considered in the work.
Sampling time was included on line 504, and light exposure data were included on line 440.
Even if the work suggest promising positive effect of ALA and SNP, the overall experiment design should be re-considered.
In the revised manuscript a new title was chosen, a hypothesis was formulated, and the scope of the manuscript was rewritten to fit with the new title.
Round 2
Reviewer 3 Report
The answers of the authors are not convincing regarding the stage of tomato development and the tween treatment effects.